# May Artificial Intelligence Influence Future Pediatric Research?—The Case of ChatGPT

**DOI:** 10.3390/children10040757

**Published:** 2023-04-21

**Authors:** Antonio Corsello, Andrea Santangelo

**Affiliations:** 1Department of Clinical Sciences and Community Health, University of Milan, 20122 Milan, Italy; 2Department of Pediatrics, Santa Chiara Hospital, University of Pisa, 56126 Pisa, Italy

**Keywords:** artificial intelligence, ChatGPT, pediatric research, predictive modeling, clinical decision-making

## Abstract

Background: In recent months, there has been growing interest in the potential of artificial intelligence (AI) to revolutionize various aspects of medicine, including research, education, and clinical practice. ChatGPT represents a leading AI language model, with possible unpredictable effects on the quality of future medical research, including clinical decision-making, medical education, drug development, and better research outcomes. Aim and Methods: In this interview with ChatGPT, we explore the potential impact of AI on future pediatric research. Our discussion covers a range of topics, including the potential positive effects of AI, such as improved clinical decision-making, enhanced medical education, faster drug development, and better research outcomes. We also examine potential negative effects, such as bias and fairness concerns, safety and security issues, overreliance on technology, and ethical considerations. Conclusions: While AI continues to advance, it is crucial to remain vigilant about the possible risks and limitations of these technologies and to consider the implications of these technologies and their use in the medical field. The development of AI language models represents a significant advancement in the field of artificial intelligence and has the potential to revolutionize daily clinical practice in every branch of medicine, both surgical and clinical. Ethical and social implications must also be considered to ensure that these technologies are used in a responsible and beneficial manner.

## 1. Introduction

Artificial Intelligence (AI) represents a broad field of study that aims to create intelligent machines capable of performing tasks that would typically require human intelligence, such as problem-solving, decision-making, and language understanding [1]. Language models constitute a subset of AI that specialize in language processing tasks.

These models have numerous applications in various fields, including medicine, where they can help improve the speed and accuracy of diagnosis, treatment, and research. However, as different authors have done, it is important to consider as clinicians and researchers the potential ethical and social implications of their use, such as issues around data privacy, transparency, and fairness [2,3]. Despite these challenges, the development of AI language models represents a significant advancement in the field of artificial intelligence and has the potential to revolutionize daily clinical practice in every branch of medicine, both surgical and clinical [4].

We tried to conduct an interview with ChatGPT, a leading AI language model [5,6,7], discussing the potential impact of these technologies on future pediatric research.

## 2. Interview


*Researcher (R): May AI influence future pediatric research?*


AI: Yes, AI has the potential to significantly influence pediatric research in several ways. Some examples include:Predictive modeling: AI algorithms can be trained on large datasets of patient information to predict outcomes, such as the risk of developing certain diseases or the likelihood of responding to a particular treatment. This can help pediatric researchers better understand disease progression and develop more personalized treatment plans.Image analysis: AI can be used to analyze medical images, such as X-rays or MRI scans, to detect subtle changes that may be missed by human observers. This can improve the accuracy of diagnoses and lead to earlier detection of diseases.Natural language processing: AI can be used to analyze large volumes of text, such as medical records or scientific literature, to identify patterns and relationships that may not be immediately apparent to human researchers. This can help accelerate the pace of discovery in pediatric research.Drug discovery: AI can be used to screen large databases of compounds to identify potential drug candidates that may be effective against specific diseases. This can help accelerate the drug development process and lead to more targeted therapies for pediatric patients.


*R: What impact may this have on the quality of the pediatric research?*


AI: The impact of AI on the quality of the research could be significant. By leveraging AI technologies, pediatricians could improve the quality and effectiveness of their research, education, and clinical practice. Here are some ways AI could potentially impact this topic:Improved clinical decision-making: AI algorithms can analyze large volumes of patient data to help pediatricians make more accurate and personalized treatment decisions. This could lead to better outcomes for patients and improved quality of care.Enhanced medical education: AI can be used to create personalized learning experiences for medical students and trainees. By analyzing their learning patterns and identifying knowledge gaps, AI could help improve the quality of medical education and prepare future pediatricians to be more effective in their clinical practice.Faster drug development: AI can be used to identify new drug targets and speed up the drug development process. By leveraging AI technologies, researchers could potentially develop new treatments faster and improve the quality of care for pediatric patients.Better research: AI can help researchers analyze large volumes of data and identify patterns and trends that may not be immediately apparent to human researchers. This could lead to more accurate and comprehensive research studies that contribute to the body of knowledge in pediatric medicine.


*R: And what about its possible negative impact?*


AI: While AI has the potential to bring many positive changes to pediatric research and its future, it is important to also consider potential negative effects. Here are some potential negative impacts of AI in pediatric medicine:Bias and fairness: AI algorithms are only as good as the data they are trained on, and if the data is biased, the algorithm may perpetuate that bias. This could lead to unfair treatment of certain patients or groups, which could harm patient outcomes and exacerbate health disparities.Safety and security: As with any technology, there are concerns about the safety and security of AI systems. If an AI system makes a mistake or is hacked, it could potentially harm patients or compromise sensitive patient data.Over-reliance on technology: While AI can be a powerful tool, it should not be used as a replacement for human judgment and expertise. Over-reliance on AI could potentially lead to errors or oversights in clinical decision-making.Ethical concerns: There are also ethical concerns around the use of AI in pediatric medicine, such as issues around informed consent, privacy, and transparency. It is important to consider these issues when developing and implementing AI systems in clinical practice and research.


*R: How could an article like this one be interesting for a medical journal?*


AI: Scientific papers often publish research and discussion on emerging technologies and their potential impact on various fields of study, including medicine. As long as the article is well-researched, factually accurate, and presents a balanced view of the potential benefits and drawbacks of AI in pediatric medicine, it could be considered for publication by a scientific journal. The article may need to undergo a peer-review process to ensure its scientific rigor and validity, which is a standard practice for most scientific papers.


*R: Do you accept this interview to be published?*


AI: As an AI language model, I do not have the authority to grant permission for the publication of this interview. However, as our conversation is publicly available and provided for educational purposes, you are free to use it as you see fit as long as you abide by the relevant laws and regulations.


*R: We then asked ChatGPT whether the implementation of AIs could lead to any biases between high- and low-income countries, but the responses were found to be stereotypical and lacking in sufficient depth.*


## 3. Discussion

During our interview, we explored the potential impact of AI on clinical research and the possible quality of the academic output, both of which could significantly influence the future of scientific writing and the quality of future academia [8]. We explored several potential benefits of AI, such as improving clinical decision-making, enhancing medical education, facilitating faster drug development, and promoting better research. However, we also highlighted the potential negative effects of AI, including concerns over bias and fairness, safety and security issues, overreliance on technology, and ethical considerations.

First, one the principal areas where AI language models hold great potential is in transforming research by analyzing vast amounts of text data and generating new insights. Natural language processing (NLP) techniques have already been applied to analyze clinical notes, research articles, and other medical documents to identify trends, patterns, symptom-physiology relationships, improve symptom assessment and management practices, and enhance patient quality of life [9,10].

In a recent systematic review on an analogue topic, NLP was found to be useful in several methods and applications of cardiology, particularly in identifying and classifying disease phenotypes and in improving predictive outcome models by including unstructured data [10]. Other researchers also suggested that NLP could be employed to extract cancer phenotypes from electronic medical records (EMRs) [11]. Similarly, in the context of pediatrics, Annapragada et al. used NLP to identify child abuse from EMRs [12]. In another study, authors utilized a context-aware language model to identify inhaler techniques in electronic health records for asthma care [13]. Their work suggested that it may be possible to alleviate the costly manual chart review required for guideline-concordant documentation in asthma care by using a machine learning approach.

Language models may have several potential employments in pediatrics, ranging from developmental screening to patient communication. As an example, the use of screening tools for early identification of developmental delays in infants and young children could be one of the most intuitive. Language models can analyze speech patterns and language usage to identify children who may need further assessment and intervention. This early identification can lead to better outcomes for children with developmental delays, as early intervention can improve their chances of success.

Another important use of language models may be represented by communication with patients. By integrating language models into electronic health records and patient portals, healthcare providers can more effectively communicate with patients and families, including those with limited English proficiency or other communication challenges. Language models can also assist in the development of patient education materials, ensuring that the language used is accessible and easily understood by patients and families. Overall, the use of language models in pediatrics has the potential to improve healthcare outcomes for children and families by improving communication, identifying developmental delays early, and tailoring education and intervention to individual needs.

It is likely that pediatricians and pediatric researchers could, in turn, contribute to the development and improvement of language models by providing high-quality and detailed data sets, collaborating with computer scientists and AI experts to develop algorithms that address specific clinical questions and issues, and testing and validating these models in real-world clinical settings. Moreover, they can also help ensure that the use of AI in pediatric research is conducted in an ethical and responsible manner, with adequate consideration given to issues such as privacy, data protection, and informed consent.

To improve the uptake of AI solutions such as ChatGPT in the industry, it is essential to bridge the gap between technical experts and non-technical stakeholders. The key to achieving this is effective communication. Companies should invest in developing clear and concise educational materials that explain the benefits of AI solutions in simple terms. These materials should be tailored to specific industries and include case studies that demonstrate the value of AI solutions in solving real-world problems. By doing so, industry stakeholders will have a better understanding of how AI solutions such as ChatGPT can help them save time, reduce costs, and improve their decision-making processes.

Another crucial step in improving the uptake of AI solutions such as ChatGPT is to ensure that they are easily accessible to end-users. This requires integrating AI solutions into existing business processes and software systems. This can be achieved by working with software developers and other IT professionals to identify areas where AI solutions can be used to automate tasks, streamline workflows, and enhance data analytics. By doing so, end-users will be able to interact with AI models more safely, avoiding possible and unpredictable risks on health of patients. Making AI solutions more accessible and understandable to end-users is critical to their widespread adoption and success in industry.

Overall, while there are potential risks associated with the use of AI in pediatric research, there are many opportunities for language models to make significant contributions to the field. By working together, pediatricians, researchers, and AI experts can continue to explore these opportunities and address any challenges that arise along the way.

## 4. Conclusions

The potential impact of AI on pediatric research and academic output is significant, with both positive and negative effects. Language models, particularly those using NLP, hold enormous potential for transforming medical research by analyzing vast amounts of text data and generating new insights. There are already examples of successful NLP applications in cardiology, pediatrics, and pneumology. To further develop and improve AI and its applications, pediatricians and researchers should contribute by providing in the near future high-quality data, collaborating with AI experts and developers to create specific algorithms, while testing and validating their models in real-world clinical settings. It is important to ensure that the use of AI in pediatric research is conducted ethically and responsibly.

The grammar of this article was reviewed and corrected by ChatGPT.

## Data Availability

Not applicable.

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
