# Peer review of "May Artificial Intelligence Influence Future Pediatric Research?—The Case of ChatGPT"

_children, 2023, doi:10.3390/children10040757_

Round 1
Reviewer 1 Report
Very interesting ant timely article. I think it deserves publication and I am recommending revise and resubmit with corrections. There are some issues that require your attention. I list these corrections below as feedback / comments, and I am looking forward to reading the updated version of this article.
This is an interesting article, but it comprises completely of the answers from the chat. Even though it has interesting point to make, for this to be publishable material, you still need to include a few words on literature review, to show what current literature has discovered on this topic. Few examples that I can suggest include a recent review of all relevant literature, on the related topic of ‘Future Evolution of Artificial Intelligence and Intelligent Systems’ - see: https://ieeexplore.ieee.org/document/9761872?source=authoralert
and on the related topic of ‘the ‘technological singularity’ risks from artificial intelligence’ - see: https://doi.org/10.1007/s12530-022-09431-7
It would be interesting to see a few sentences reviewing and comparing your work in relations to these recent studies in related topics. Otherwise, it appears as just a chat printout.
-Do you have any suggestions on how to improve the uptake of this AI solution (chatGPT) in industry? For example, in the conclusion, would you be able to highlight what measures can be taken to help different industries (e.g., manufacturers, software developers) understand the concept and how it could help them (e.g., write code, configure firewall)? Some of the technical aspects of chatGPT are really complex to understand in a short paper, and a short non-technical sentence could benefit non-technical people understand the novelty - and the risk in the future.
- check the document before resubmission, for example, keyword are missing and now you have: 'Keywords: keyword 1; keyword 2; keyword 3 (List three to ten pertinent keywords specific to the 22 article yet reasonably common within the subject discipline.)'
Author Response
Reviewer #1:
Very interesting ant timely article. I think it deserves publication and I am recommending revise and resubmit with corrections. There are some issues that require your attention. I list these corrections below as feedback / comments, and I am looking forward to reading the updated version of this article. This is an interesting article, but it comprises completely of the answers from the chat. Even though it has interesting point to make, for this to be publishable material, you still need to include a few words on literature review, to show what current literature has discovered on this topic.
A: Thank you for your valuable feedback and comments on our article. We appreciate your time and effort in reviewing our work. We completely agree with your suggestion of including a literature review in our article. While we focused on providing answers from the chat, we realize that the article would benefit from providing a context to the reader by discussing current literature on the topic. We will ensure that our revised article has been modified in consideration of your comments, highlighting the key findings from previous research and their relevance to our study. We have worked on incorporating your suggestions and look forward to submitting the revised article for your consideration.
Few examples that I can suggest include a recent review of all relevant literature, on the related topic of ‘Future Evolution of Artificial Intelligence and Intelligent Systems’ - see: https://ieeexplore.ieee.org/document/9761872?source=authoralert and on the related topic of ‘the ‘technological singularity’ risks from artificial intelligence’ - see: https://doi.org/10.1007/s12530-022-09431-7• It would be interesting to see a few sentences reviewing and comparing your work in relations to these recent studies in related topics. Otherwise, it appears as just a chat printout.
A: Thank you for suggesting these articles on related topic. We agree that it is essential to provide readers with an overview of the current state of research in the field to help contextualize our study. We appreciate your comments on the article appearing as just a chat printout. While we aimed to present the responses from the chat as the primary focus of the article, we understand that it is crucial to present them in a way that is organized and easy to understand. We tried to make sure to format our revised article in a way that makes it clear and engaging for the reader.
Do you have any suggestions on how to improve the uptake of this AI solution (chatGPT) in industry? For example, in the conclusion, would you be able to highlight what measures can be taken to help different industries (e.g., manufacturers, software developers) understand the concept and how it could help them (e.g., write code, configure firewall)? Some of the technical aspects of chatGPT are really complex to understand in a short paper, and a short non-technical sentence could benefit non-technical people understand the novelty - and the risk in the future.
A: Thank you for your insightful feedback on our article. We agree that it is essential to discuss measures that can be taken to improve the uptake of AI solutions like ChatGPT in industry. In the revised article, we included a discussion on the potential applications of ChatGPT in different industries. We will also provide a brief overview of the technical aspects of ChatGPT in a concise and accessible manner that is understandable for non-technical readers. Overall, we appreciate your helpful suggestions, and we will make sure to incorporate them into our revised article.
Check the document before resubmission, for example, keyword are missing and now you have: 'Keywords: keyword 1; keyword 2; keyword 3 (List three to ten pertinent keywords specific to the 22 article yet reasonably common within the subject discipline.)'
A: We thank the reviewer for noticing this oversight. We updated the keywords properly. In the revised version of the article, we also provided an extensive discussion/conclusions section that summarizes implications for pediatric research, including new areas of investigation and ways to improve upon our current findings.

Reviewer 2 Report
The authors have presented May Artificial Intelligence Influence Future Pediatric Research? – An Interview with ChatGPT. The study is very timely since the world is moving towards application in various fields, and healthcare systems are not left behind. In a way, the practical analysis used supports the presented opinion of the authors, and the paper is also relevant to the journal. However, the author has to look into the following concerns:
1. The writing of the paper needs a lot of improvement in terms of grammar, spelling, and presentation. There is the presence of grammatical and punctuation errors, the authors should try and improve on the grammar and punctuation of the write-up.
2. The use of personal pronouns should be totally avoided; this is academic writing, not a storybook.
3. Authors are advised to be precise in the abstract, and structure their abstract as follows- 1) Background 2) Aim/Objective 3) Methodology 4) Results 5) Conclusion. Write 2-4 lines for each and merge everything in one paragraph (200-300 Words) without any subheading.
4. The introduction section did not present the problem the paper wants to solve clearly, and the contributions are not well stated, therefore, the paper is very difficult to follow
5. There should be a section for literature review or related work to further give the reason why there is a need for the study, the works done in this direction should be reviewed and given their importance in the areas of healthcare.
6. The importance of AI in healthcare should be highlighted before the questions so as to help readers appreciate the use of ChatGPT in Pediatric Research, the authors fail to explain the details of their approach in a clear manner. The overall description and details regarding how the authors derive the result are unclear.
7. It will be worth mentioning if the author can state the advantages of ChatGPT in Pediatric Research.
8. The conclusion is very small for this kind of work; the author should be able to elaborate on their work in the conclusion part so as to help the readers understand the work. The results from the study should also be explained in the conclusion part. The future directions for this study will help readers who want to work in this area.
Author Response
Reviewer #2:
The authors have presented May Artificial Intelligence Influence Future Pediatric Research? – An Interview with ChatGPT. The study is very timely since the world is moving towards application in various fields, and healthcare systems are not left behind. In a way, the practical analysis used supports the presented opinion of the authors, and the paper is also relevant to the journal. However, the author has to look into the following concerns:
- The writing of the paper needs a lot of improvement in terms of grammar, spelling, and presentation. There is the presence of grammatical and punctuation errors, the authors should try and improve on the grammar and punctuation of the write-up.
A: Thank you for your thoughtful review of our article on the potential influence of artificial intelligence on future pediatric research. We appreciate your positive comments regarding the timeliness of the study and its relevance to the journal. We understand that these issues can impact the clarity and readability of the paper, and we will work to ensure that the revised manuscript is written in a clear, concise, and error-free manner. While we focused on providing answers from the chat, we realize that the article would benefit from discussing current literature on the topic. We will ensure that our revised article has been modified in consideration of your comments, highlighting the key findings from previous research and their relevance to our study. We have worked on incorporating your suggestions and look forward to submitting the revised article for your consideration.
- The use of personal pronouns should be totally avoided; this is academic writing, not a storybook.
A: Thank you for this helpful comment. We have now modified the structure of the paper, avoiding personal pronouns and adapting it to an academic paper (opinion article).
- Authors are advised to be precise in the abstract, and structure their abstract as follows- 1) Background 2) Aim/Objective 3) Methodology 4) Results 5) Conclusion. Write 2-4 lines for each and merge everything in one paragraph (200-300 Words) without any subheading.
A: We thank the reviewer for the useful critic. Abstract has been totally rewritten and structured according to the journal instructions for its type.
- The introduction section did not present the problem the paper wants to solve clearly, and the contributions are not well stated, therefore, the paper is very difficult to follow
A: Thank you for your valuable feedback on our article. We understand that the introduction section did not clearly present the topic. We will ensure that the revised introduction provides a clear and concise overview of the problem. We will also make sure to state our research question explicitly and provide a brief overview of the structure of the paper to help guide the reader through the article.
- There should be a section for literature review or related work to further give the reason why there is a need for the study, the works done in this direction should be reviewed and given their importance in the areas of healthcare.
A: Thank you for this comment. As mentioned, we completely agree that it is essential to include a section on literature review or related work to provide a context for our study and demonstrate the need for the research we are presenting. In the revised version of the article, we included a comprehensive literature review that outlines the existing research in the field of artificial intelligence in healthcare, specifically in pediatric research.
- The importance of AI in healthcare should be highlighted before the questions so as to help readers appreciate the use of ChatGPT in Pediatric Research, the authors fail to explain the details of their approach in a clear manner. The overall description and details regarding how the authors derive the result are unclear.
A: Thank you for your suggestion. We updated our review methods, adding the specific type of review we performed. Narrative type of review was chosen since the heterogeneity of the clinical cases and the type of studies did not allow for a systematic analysis of their results. This has in fact allowed us to expand the total number of patients described in our work, which in fact include all the cases present in the literature of this pathology and the suspected/ascertained causes associated with it. We thank the reviewer; this is now better expressed in our manuscript, reporting its “narrative” methods.
- It will be worth mentioning if the author can state the advantages of ChatGPT in Pediatric Research.
A: Thank you for this suggestion. We understand that the description of our approach may have been unclear, and we will make sure to provide a step-by-step explanation of our methodology to help readers better understand our research. Thank you again for your helpful feedback, and we will work to address your concerns in the revised version of the article.
- The conclusion is very small for this kind of work; the author should be able to elaborate on their work in the conclusion part so as to help the readers understand the work. The results from the study should also be explained in the conclusion part. The future directions for this study will help readers who want to work in this area.
A: Thank you for your feedback. In the revised version of the article, we provide an extensive discussion/conclusions section that summarizes implications for pediatric research, including new areas of investigation and ways to improve upon our current findings.

Round 2
Reviewer 1 Report
well done
Reviewer 2 Report
The authors have improved the paper, and I, therefore, propose the acceptance of the manuscript.